# Removal of Apixaban during Emergency Cardiac Surgery Using Hemoadsorption with a Porous Polymer Bead Sorbent

**DOI:** 10.3390/jcm11195889

**Published:** 2022-10-05

**Authors:** Kambiz Hassan, Matthias Thielmann, Jerry Easo, Markus Kamler, Daniel Wendt, Zaki Haidari, Efthymios Deliargyris, Mohamed El Gabry, Arjang Ruhparwar, Stephan Geidel, Michael Schmoeckel

**Affiliations:** 1Department of Cardiac Surgery, Asklepios Klinik St. Georg, 20099 Hamburg, Germany; 2Department of Thoracic- and Cardiovascular Surgery, West-German Heart and Vascular Center, University Duisburg-Essen, 45147 Essen, Germany; 3Department of Cardiac Surgery Essen-Huttrop, University of Essen, 45138 Essen, Germany; 4CytoSorbents, Princeton, NJ 08540, USA

**Keywords:** apixaban, blood purification, DOAC, hemoadsorption, porous polymer beads, ‘CytoSorb’

## Abstract

Background: Patients on direct oral anticoagulants are at high risk of perioperative bleeding complications. We analyzed the results of intraoperative hemoadsorption (HA) in patients undergoing cardiac surgery who were also on concurrent therapy with apixaban. Methods: we included 25 consecutive patients on apixaban who underwent cardiac surgery with the use of cardio-pulmonary bypass (CPB) at three sites. The first 12 patients underwent surgery without hemoadsorption (controls), while the next 13 consecutive patients were operated with the Cytosorb^®^ (Princeton, NJ, USA) device integrated into the CPB circuit (HA group). The primary outcome was perioperative bleeding assessed by the Bleeding Academic Research Consortium (BARC) definition and secondary outcomes included 24 h chest-tube-drainage (CTD) and need for 1-deamino-8-d-arginine-vasopressin (desmopressin (DDAVP)) administration to achieve hemostasis. Results: Preoperative mean daily dose of apixaban was higher in the HA group (8.5 ± 2.4 vs. 5.6 ± 2.2 mg, *p* = 0.005), while time since last apixaban dose was longer in the controls (1.3 ± 0.9 vs. 0.6 ± 1.2 days, *p* < 0.001). No BARC-4 bleeding events and no repeat-thoracotomies occurred in the HA group compared with 3 and 1, respectively, in the controls. Postoperative 24 h CTD volume was significantly lower in the HA group (510 ± 152 vs. 893 ± 579 mL, *p* = 0.03) and there was no need for DDAVP compared to controls, who received an average of 10 ± 13.6 mg (*p* = 0.01). Conclusions: In patients on apixaban undergoing emergent cardiac surgery, the intraoperative use of hemoadsorption was feasible and safe. Compared to patients operated on without hemoadsorption, BARC-4 bleeding complications did not occur and the need for 24 h CTD and DDAVP was significantly lower.

## 1. Introduction

Direct-acting oral anticoagulants (DOACs) are now the preferred agents for long-term anticoagulation because they are more predictable and have a better benefit: risk profile than warfarin [1]. Apixaban and rivaroxaban are by far the most commonly prescribed DOACs, primarily for stroke prevention in patients with non-valvular atrial fibrillation (AF), and have emerged as the preferred choice, particularly in patients new to oral anticoagulant therapy [2]. Accordingly, there is an increase in the number of patients receiving DOACs and undergoing cardiac surgery, which carries an increased risk of perioperative bleeding, especially when the surgery is urgent. In cases where surgery can be postponed, the recommended preoperative discontinuation period (“washout”) is at least 48 h and is even longer in patients with impaired renal function [3].

In the cases when surgery takes place before the recommended washout period, severe bleeding complications are frequent and, to date, treatment is limited to non-specific, supportive therapies that include blood product transfusions (platelets, red blood cells, etc.) and administration of coagulation factors [3]. Therefore, alternate strategies are urgently needed to improve the safety of patients on anticoagulants undergoing non-deferrable surgery with high perioperative bleeding risk. Recently, apixaban removal by hemoadsorption has been demonstrated in vitro [3,4] and in a single case report [5]. As a result, intraoperative hemoadsorption may be a novel approach to reduce the risk of perioperative bleeding in patients undergoing emergent or urgent cardiac surgery. On the other hand, new costly reversal agents (idarucizumab targeting dabigatran and andexanet alfa targeting rivaroxaban and apixaban) are currently investigated.

We hypothesized that the intraoperative use of CytoSorb^®^ could be used to remove apixaban from whole blood, mitigating postoperative bleeding complications defined by BARC-4 criteria. The present study is the first clinical experience with the intraoperative use of Cytosorb^®^ hemoadsorption for apixaban removal at three large cardiac surgery centers and evaluates the impact on bleeding outcomes in this high-risk population.

## 2. Materials and Methods

### 2.1. Patients

The present study was a non-randomized, multi-center prospective study of 25 consecutive patients on apixaban medication who underwent cardiac surgery either at the Department of Cardiac Surgery at the Asklepios Heart Center Hamburg, the West-German Heart and Vascular Center Essen or the Department of Cardiac Surgery in Essen-Huttrop, between January 2017 and June 2022. All patients were on apixaban medication and all types of surgeries (except aortic cases) were included (all with aortic cross clamping). Intraoperatively, in all patients an antifibrinolytic strategy with tranexamic acid was applied. Other inclusion criteria were: use of CytoSorb^®^ for antithrombotic removal in the HA group and informed consent for prospective registry participation. Exclusion criteria were washout periods > 48 h, use of CytoSorb^®^ for purposes other than antithrombotic removal, pregnancy, life expectancy < 1 year (potential cancer as an underlying disease) and age < 18 years. In addition to this, no patients presenting with aortic disease were included in the present analysis. The present study obtained Institutional Review Board approval according to the Declaration of Helsinki by the Hamburg General Medical Council Ethics Committee and by the local central Ethics Committee in Essen.

The first 12 patients underwent surgery without hemoadsorption and with a withdrawal interval of at least 24 h and at most 48 h before surgery (controls), whereas the next 13 consecutive patients underwent surgery with the Cytosorb^®^ device integrated into the cardiopulmonary bypass (CPB) circuit (HA group). EuroSCORE II was calculated using an online calculator (https://www.euroscore.org/index.php?id=17 (accessed on 29 August 2022)).

### 2.2. Bleeding Classification

Bleeding complications were classified according to the BARC criteria (Bleeding Academic Research Consortium). BARC type 4 was of interest for the present analysis and served as the primary outcome parameter, because it specifically refers to bleeding associated with coronary artery bypass grafting (CABG) within 48 h. BARC type 4 includes: perioperative intracranial bleeding within 48 h or re-operation after closure of sternotomy for the purpose of controlling bleeding or transfusion of 5 units of whole blood or packed red blood cells within 48 h or chest tube output of 2000 mL within 24 h.

### 2.3. Device

The CytoSorb^®^ 300 mL device (CytoSorbents, Princeton, NJ, USA) is filled with highly biocompatible, porous polymer beads covered with a divinylbenzene coating. Each polymer bead is between 300 and 800 μm in size and has pores and channels, resulting in an effective surface area of more than 40,000 m^2^ capable of binding hydrophobic small and medium-sized molecules [6]. Cytosorb^®^ is CE marked according to the Medical Devices Directive (ISO 10,993 biocompatible, manufactured in the United States under ISO 13,485 certification). The adsorber was integrated in the CPB circuit between the oxygenator and the venous reservoir as previously described by our group [7,8]. Figure 1 exemplifies the intraoperative setup.

### 2.4. Statistical Analysis

Data were analyzed using STATA/BE version 17 software (StataCorp LLC, College Station, TX, USA). Continuous variables were expressed as mean ± standard deviation (SD) and median and interquartile range and compared using Student’s *t*-test or the Mann–Whitney test. Categorical data were expressed as number of patients and frequencies and compared with the chi-square test. *p*-values < 0.05 were considered significant.

## 3. Results

### 3.1. Baseline Characteristics

From January 2017 to June 2022, a total of 25 patients on apixaban underwent on-pump cardiac surgery at the three participating centers. The first 12 patients did not receive intraoperative hemoadsorption (controls), whereas the subsequent 13 patients did (HA group). The groups differed with respect to the time interval since last apixaban intake (1.3 ± 0.9 vs. 0.6 ± 1.2 days, *p* < 0.001), with a longer discontinuation time in the control group. In addition, the mean apixaban dose was significantly higher in the HA group (8.5 ± 2.4 vs. 5.6 ± 2.2 mg, *p* = 0.005). Patients with known coronary heart disease were also treated preoperatively with a daily dose of 100 mg aspirin. Preoperative baseline characteristics of the patients are displayed in detail in Table 1. The baseline EuroSCORE II was numerically higher in the control group than in the HA group (7.4 ± 8.0 vs. 4.4 ± 1.8%, *p* = 0.25) without being statistically significant.

### 3.2. Operative Outcomes

The majority of patients underwent a CABG procedure, either isolated or combined with mitral valve repair, left atrial appendage closure and/or a Maze procedure. In addition, bilateral mammary artery was used in more than 50% of the patients in both groups. Mean aortic cross clamp (ACC) time was comparable between the two groups (80.9 ± 22.3 in the HA group vs. 80.9 ± 27.8 min in the control group, *p* = 0.99). CPB time was also not significantly different, with slightly longer times in the HA group (119.7 ± 30.5 vs. 109.1 ± 28.1 min, *p* = 0.37). The total operation time (skin-to-skin) was shorter in the HA group without reaching statistical significance (279.8 ± 56.0 vs. 305.2 ± 76.9 min, *p* = 0.35). However, regarding hemostasis, the dry-up time (skin-to-skin time subtracted by CPB time) was lower in the HA group (160.1 ± 53.4 vs. 196.1 ± 54.7 min, *p* = 0.11) without being significantly different. Serious perioperative bleeding events (BARC-4) were observed only in the control group (4 vs. 0, *p* = 0.09), including one patient who required a repeat-thoracotomy due to bleeding, one patient who required more than 5 RBC units and two patients who had a chest tube output of more than 2000 mL (one of these two received more than 5 RBC). Postoperative 24 h CTD was significantly lower after HA (510 ± 152 vs. 893 ± 579 mL, *p* = 0.03) and is displayed in Figure 2. Control patients received an average of 10 ± 13.6 mg of 1-deamino-8-d-arginine vasopressin (desmopressin (DDAVP)) to achieve hemostasis, whereas DDAVP was not administered in any patients in the HA group (*p* = 0.01). Operative characteristics and postoperative outcome results are presented in Table 2.

## 4. Discussion

This present study investigated perioperative bleeding complications in patients on apixaban medication undergoing non-deferrable on-pump cardiac surgery, with or without intraoperative hemoadsorption. The main observation from the current study is: intraoperative hemoadsorption was associated with less perioperative bleeding, as evidenced by a significant reduction in 24 h CTD and no BARC-4 bleeding events (as the primary outcome), which are otherwise common in this high-risk population. Importantly, these results were observed despite higher apixaban dosing and shorter washout times compared with patients without hemoadsorption.

In the current study population, it should be noted that only urgent operations were included. Even in this setting, in a high percentage of patients, a bilateral mammary artery approach, as well as occlusion of the left atrial appendage, was performed. This is in line with the latest results of the LAAOS III trial [9]. Notably, aortic cross-clamping was similar between both groups, whereas the CPB times were slightly longer in the adsorber group, without reaching statistical significance. Regarding the total operation time, there was no significant difference between both groups. Nevertheless, the absolute difference, although not significant, between both groups showed a remarkable reduction in total operation time of almost 25 min in the adsorber group. One could speculate that the time required to achieve hemostasis might be less when intraoperative hemoadsorption is used. In addition to this, once the CPB time was subtracted from the overall operation time, an insignificant lower dry-up time for the HA group was observed. Moreover, the therapy was safe (no device-related adverse events occurred) and the adsorber could be easily integrated into the CPB circuit. So far, no other complications have been reported without any other side-effects. In addition to this, the intraoperative heparin regime must not be changed, as the therapy is proven and safe and no heparin removal was shown.

Cardiac surgery itself is a major surgical procedure and, therefore, carries a high risk of perioperative bleeding, even in patients not on antithrombotic therapy. However, antithrombotic drugs (P2Y12 inhibitors, DOACs, etc.) further increase the risk of bleeding, especially in patients who require urgent or emergent major surgery without an adequate washout period. In the non-cardiac surgical setting, it was shown by the PAUSE trial that stopping DOACs for 1 to 4 days before surgery is safe (one day for a low-risk and 2 days for a high-risk procedure), with prolonged discontinuation necessary in patients with impaired renal function [10]. However, for cardiac surgery, this interval should be extended to 4 to 5 days, corresponding to an elimination half-life of 10 days [10,11,12], especially when drug level monitoring or pharmacodynamic tests specific to DOACs are not available. One observational study even suggests that bleeding risk may extend beyond 4 days and that longer washout periods may be required.

DOACs, as non-vitamin-K antagonists, are today represented by four drugs: the direct activated factor II (thrombin) inhibitor dabigatran and three activated factor X (FXa) inhibitors—apixaban, rivaroxaban and edoxaban. Apixaban and rivaroxaban are by far the most frequently prescribed DOACs worldwide, mostly for the prevention of stroke in patients with non-valvular atrial fibrillation, and have emerged as the preferred choice, particularly for those newly started on oral anticoagulant therapy [2,13]. This is based on the fact that DOACs are easier to manage for the patient and have a fast onset and offset, which is, however, not fast enough in the setting of urgent or emergent in particular cardiac surgery.

In the emergency setting, idarucizumab and andexanet alfa, which reverse the anticoagulant effects of dabigatran and FXa inhibitors, respectively, are DOAC reversal agents. These new drugs come with extremely fast and almost 100% reversal; however, no data in cardiac surgery for andexanet alfa, especially in regard to interactions with heparin, are known. They also come at a very high cost (Andexanet alfa USD 24,750, incompatible with heparin and potential rebound effect). Due to cost effectiveness, intraoperative hemoadsorption has been recently recommended by the National Institute for Health and Care Excellence in the UK. Moreover, Javanbakht et al. could show a cost reduction in a UK-based cost-utility analysis of CytoSorb^®^ in patients treated with ticagrelor [14]. Their analysis revealed that the treatment with CytoSorb^®^ could save almost GBP 4000 over a 30-day time horizon due to consumption of less blood products, fewer re-thoracotomies and shorter length of stay in the emergency surgical setting, with the price of the device taken into account [2,14].

In the past, we studied outcomes and bleeding complications after use of DOACs in patients undergoing cardiac surgery [15]. As a next step, we showed that Cytosorb^®^ adsorption during emergency cardiac surgery in patients at high risk of bleeding (pretreated with ticagrelor or rivaroxaban) was an effective method to reduce bleeding complications and should be used routinely in such patients [7]. This could be also verified by our group in the acute setting of aortic dissections where intraoperative hemoadsorption was capable of attenuating bleeding risk in patients pretreated with ticagrelor or rivaroxaban [8]. Just recently, a group from Oslo demonstrated the in vitro removal of apixaban from whole blood using CytoSorb^®^: within 30 min of adsorption, the mean apixaban concentration was reduced from 414.3 (±69.1) ng/mL down to 33 (±11.4) ng/mL and reversal of the anticoagulant effect with normalized hemostasis [4]. Systematic, benchtop testing using a “life-size” recirculation model designed to mimic conditions encountered in clinical, intraoperative use also demonstrated efficient removal of apixaban, rivaroxaban and ticagrelor from whole blood with the DrugSorbTM-AntiThrombotic Removal (ATR) hemoadsorption device (CytoSorbents Inc., Princeton, NJ, USA) [3]. It is, therefore, reasonable to presume that observed in vitro drug removal capabilities are also translatable to the in vivo setting. In the current study, we sought to evaluate real-world clinical data with the intraoperative use of Cytosorb^®^ in patients on apixaban undergoing non-deferrable on-pump cardiac surgery.

The current results extend the clinical evidence for intraoperative apixaban removal that was previously limited to a single case report [5]. The oral dosing of apixaban varies according to indication, but 5 mg twice daily is most commonly prescribed. In the present analysis, the hemoadsorption group was treated with a significantly higher dose of apixaban preoperatively. Moreover, and probably even more clinically important, the HA group was operated sooner after the last apixaban dose without any meaningful washout of apixaban. Nearly all patients in the present study received apixaban due to non-valvular atrial fibrillation and the vast majority of patients presented with an urgent or emergent indication for CABG surgery.

In regard to postoperative bleeding events, three patients from the control group showed BARC type 4 bleeding, with a total of four BARC-4 events as the primary outcome parameter. Moreover, the hemoadsorption group experienced a significantly lower chest tube loss over 24 h compared to the control group. The current literature supports 24 h chest tube drainage as a clinically meaningful metric, with well-established prognostic value, including a known linear relationship with in-hospital mortality [16]. Relationships between 24 h CTD and transfusion volume, length of ICU stay, length of hospital stay and increased need for supportive ventilation and/or renal replacement therapy have also been demonstrated [17]. Indeed, established precedence exists for use of 24 h CTD as primary outcomes in cardiac surgical studies [18,19,20]. Additionally, our study population is on apixaban, which contributes to both greater and more prolonged chest tube blood loss, so the assessment of CTD out to 24 h is especially relevant and important in our population. Of note, a post hoc power calculation (power 0.8, α-level of 0.05, two-sided *t*-test and 12 patients per group, 24 in total) resulted in a Cohen’s d co-efficient of 1.2. In other words, this implies a strong effect of the independent factor (hemoadsorption), resulting in a clinically measurable effect.

In conclusion, this is, to the best of our knowledge, the first study to show that intraoperative hemoadsorption has the potential to improve outcomes in patients on apixaban undergoing nondeferrable on-pump cardiac surgery by mitigating the risk of perioperative bleeding complications. Moreover, it was shown in the past that hemoadsorption is also capable of removing ticagrelor from blood and is highly cost effective compared to expensive reversal agents.

Our study has two significant limitations that should be considered when interpreting the results. First, the small sample size and, second, the non-randomized nature of the allocation of patients in the two groups cannot exclude the possibility of unknown confounders influencing the results. Comorbidities, such as renal or hepatic impairment, could influence the natural washout of apixaban. In the present analysis, however, these comorbidities were not reported in either group and, therefore, the slower, natural elimination would not be expected. Moreover, the consecutive nature in treatment allocation (controls first and HA group later) suggests that selection bias for who would receive HA was not at play. Further, the directional consistency across all the outcomes examined (BARC-4 events as the primary outcome parameter, 24 h CTD, DDAVP use) argues against the risk for high random bias underlying the results. Despite the aforementioned limitations, it is important to note that the current study represents the largest evaluation of hemoadsorption for intraoperative apixaban removal to date.

## Figures and Tables

**Figure 1 jcm-11-05889-f001:**
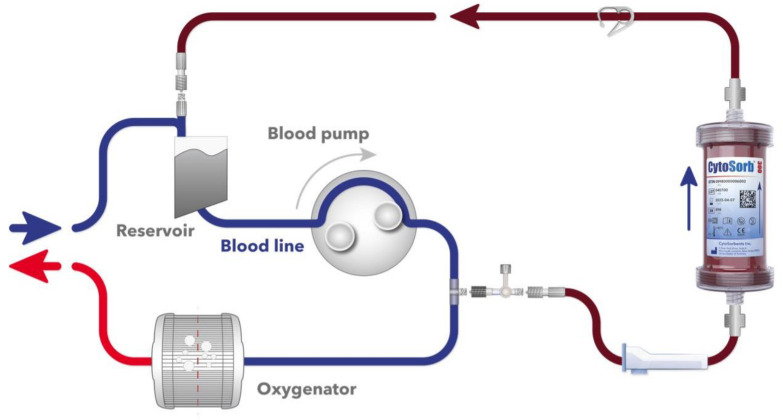
Schematic of the adsorber incorporated into the CPB circuit.

**Figure 2 jcm-11-05889-f002:**
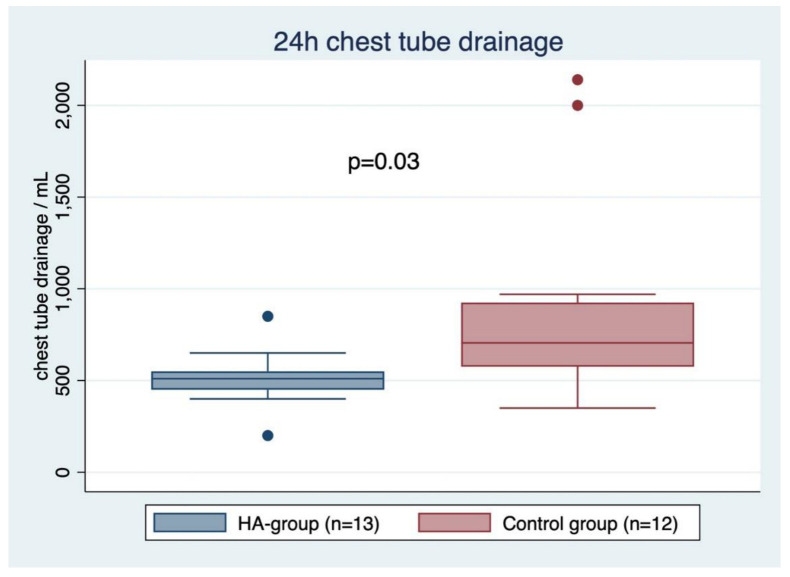
Box plot of chest tube drainage (including minimum, first quartile, median, third quartile and maximum).

**Table 1 jcm-11-05889-t001:** Baseline characteristics.

Variable	HA Group n= 13	Control Group n = 12	*p*
**Demographics**			
Age, years	67.7 ± 9.4 69 [61.5, 72.5]	71.2 ± 6.5 72.5 (69, 75)	0.29
Gender, male	11 (84.6)	9 (75.0)	0.64
BMI, m^2^	30.0 ± 5.4 32 (26, 32.5)	27.4 ± 2.9 28 (25.8, 29.4)	0.14
COPD	3 (23.1)	0 (0.0)	0.22
Systemic hypertension	13 (100.0)	10 (83.3)	0.22
Smoking	4 (15.4)	2 (16.7)	0.64
Hyperlipidemia	9 (69.2)	6 (50.0)	0.43
Diabetes	6 (46.2)	7 (58.3)	0.69
Atrial fibrillation	12 (92.3)	12 (100.0)	1.00
Peripheral vascular disease	3 (23.1)	3 (25.0)	1.00
GFR, ml/min/1.73 m^2^	77.6 ± 17.9 80 (70, 86.8)	67.2 ± 33.6 77 (39.8, 86.8)	0.33
Reoperation	1 (20)	0 (0.0)	1.00
Ejection fraction, %	49.7 ± 8.1 50 (45, 56)	49.8 ± 10.4 50 (43.3, 56.3)	0.96
EuroSCORE II, %	4.4 ± 1.8 5 (3.2, 5.6)	7.4 ± 8.0 5.4 (3.7, 8.2)	0.25
Daily dose of apixaban, mg	8.5 ± 2.4 10 (5, 10)	5.6 ± 2.2 5 (5, 5)	0.005
Withdrawal interval, days	0.6 ± 1.2 0 (0, 1)	1.3 ± 0.9 1 (0.8, 1.5)	<0.001

Data are presented as mean ± SD and median and IQR; BMI: body mass index; COPD: Chronic obstructive pulmonary disease; GFR: Glomerular filtration rate.

**Table 2 jcm-11-05889-t002:** Operative and postoperative characteristics.

Variable	HA Group n = 13	Control Group n = 12	*p*
**Operative characteristics**			
ACC, min	80.9 ± 22.3 76 (67, 105)	80.9 ± 27.8 81.5 (71, 88.3)	0.99
CPB, min	119.7 ± 30.5 124 (104, 135)	109.1 ± 28.1 112 (92, 128)	0.37
Total OR time, min	279.8 ± 56.0 269 (237, 312)	305.2 ± 76.9 305 (264, 349)	0.35
Isolated CABG	3 (23.1)	3 (25.0)	1.00
CABG + LAA occlusion	3 (23.1)	5 (41.2)	0.41
CABG + LAA occlusion + Maze	6 (46.2)	4 (33.3)	0.69
CABG + MVR	0 (0.0)	1 (8.3)	1.00
BIMA use	7 (53.8)	8 (66.7)	0.68
MVRe	1 (7.7)	0 (0.0)	1.00
**Perioperative outcomes**			
Rethoracotomy	0 (0.0)	1 (8.3)	0.48
>5 RBC/48 h	0 (0.0)	1 (8.3)	0.48
DDAVP, mg	0 (0.0) 0 (0, 0)	10 ± 13.6 0 (0, 25)	0.01
Drainage volume/24 h, mL	510 ± 152 510 (450, 550)	893 ± 579 705 (588, 902)	0.03
>2000 mL/24 h	0 (0.0)	2 (16.7)	0.22
Intracranial bleeding	0 (0.0)	0 (0.0)	1.00
BARC 4 bleeding	0 (0.0)	3 (25.0)	0.09
ICU stay, days	3.1 ± 2.4 2 (2, 3)	3.7 ± 3.2 2.5 [1.8, 4]	0.60
Hospital stay, days	13.0 ± 4.7 13 (11, 150)	14.7 ± 11.5 12 (11, 14)	0.63
In-hospital mortality	0 (0.0)	1 (8.3)	0.48

Data are presented as mean ± SD and median and IQR; ACC: Aortic cross clamping; CPB: Cardio-pulmonary bypass; LAA: Left atrial appendage; MVR: Mitral valve repair; BIMA: Bilateral internal mammary artery; MVRe: Mitral valve replacement; DDAVP: 1-deamino-8-d-arginine vasopressin; BARC: Bleeding Academic Research Consortium.

## Data Availability

Not applicable.

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
