# Peer review of "Removal of Apixaban during Emergency Cardiac Surgery Using Hemoadsorption with a Porous Polymer Bead Sorbent"

_jcm, 2022, doi:10.3390/jcm11195889_

Round 1

Reviewer 1 Report

we thank the reviewer for this mlanuscript describing an historical cohort of 25 patients receiving pre operative Apixaban and analysing the strategy of removal during CPB. 

The innovation presented here is remarquable, nevertheless the manuscript must be improved regarding several critical points: 

1. the main hypothesis must be clearly presented. 

2. a main outcome must be clearly defined, and used all along the project. Here, the BARC type 4 seems to be the most relevant. 

3. hence, a power evaluation , even post hoc, should be evocated.

4. For the results, regardly the number of patient, the quantitatives variables should be presented with median and interquartile range. 

4.bis.   Did the author collected pre operative biological data such as haemostasis lab nor point of care thromboelastometry? No information regarding the Fibrinogen level are presented ( during the pre and peri operative period)

5. As the BARC type 4 score is used as main outcome, it must be presented in priority, no statistical test should be performed upon the different variable of the composite score ( as describe in the introduction). This could limit the "cherry picking" of p-value... 

6. The authors must provide the anti-fibrinolytic strategy used during the study period as it could influence the main outcome... 

7. only 2 patients presented a massive bleeding, it could be interesting to explain why. Did they have any clinical specificity ? 

8. Idarucizumab and andexanet alfa are presented in the discussion, should they be discussed in the introduction also ? 

9. It could be pertinent to add a flow shart and discuss the exclusion criteria of "life expectancy <1 year", did you study team had Aortic surgery among patients with DOAC excluded ? 

Author Response

Reviewer: 1

Comments to the Author

We thank the reviewer for this manuscript describing an historical cohort of 25 patients receiving pre operative Apixaban and analyzing the strategy of removal during CPB.

The innovation presented here is remarkable, nevertheless the manuscript must be improved regarding several critical points:

Question

  1. the main hypothesis must be clearly presented.

Answer

We are thankful for this remark of the reviewer.  We have now added the hypothesis into the revised introduction part of the manuscript. 

Therefore, the following has been added into the revised introduction part:

“We hypothesized that the intraoperative use of CytoSorb® could be used to remove apixaban from whole blood mitigating postoperative bleeding complications defined by BARC-4 criteria.”

Question

  1. a main outcome must be clearly defined, and used all along the project. Here, the BARC type 4 seems to be the most relevant.

Answer

We greatly appreciated this well-taken remark.  Of course, BARC-4 bleeding definition was used as the primary endpoint parameter and all others were defined as secondary outcomes.  This has now been clarified, starting in the abstract and used throughout the manuscript.

Added to the revised version of the abstract:

“The primary outcome was perioperative bleeding assessed by the Bleeding Academic Research Consortium (BARC) definition and secondary outcomes included 24-hour chest-tube-drainage (CTD) and need for 1-deamino-8-d-arginine-vasopressin (desmopressin [DDAVP]) administration to achieve hemostasis.”

Question

  1. hence, a power evaluation, even post hoc, should be evocated.

Answer

We appreciate this interesting point.  Although the current paper was not designed as a prospective trial, we now did the requested post-hoc power calculation.  We added the following into the revised discussion part:

“Of note, a post-hoc power calculation (power 0.8, a-level of 0.05, 2-sided t-test and 12 patients per group, 24 in total) resulted in a Cohen´s d co-efficient of 1.2.  In other words, this implies a strong effect of the independent factor (hemoadsorption) resulting in a clinical measurable effect.”

Question

  1. For the results, regardly the number of patient, the quantitatives variables should be presented with median and interquartile range.

Answer

According to the reviewers suggestion, we also added for quantitative variables the median and interquartile range in addition to the mean and standard deviation to all tables.

Question

4.bis.   Did the author collected pre operative biological data such as haemostasis lab nor point of care thromboelastometry? No information regarding the Fibrinogen level are presented ( during the pre and peri operative period)

Answer

We appreciate this remark.  However, no additional hemostasis parameters and no Point-of-Care testing such as ROTEM, VerifyNow or Multiplate was performed.  Therefore, we cannot present additional data on this topic.  For a future prospective trial, we will definitively consider this interesting and important topic.  The current trial was intended to be a pilot study on removing apixaban.

Question

  1. As the BARC type 4 score is used as main outcome, it must be presented in priority, no statistical test should be performed upon the different variable of the composite score ( as describe in the introduction). This could limit the "cherry picking" of p-value...

Answer

We are thankful for this remark.  We definitively did not intend to perform any cherry picking of any p-values, but rather want to present the BARC-4 score and to be more precise and reliable all sub-values, as this could be of potential remark.  Nevertheless, we have now put more emphasis on the BARC-4 classification throughout the whole manuscript as the primary outcome parameter.

Question

  1. The authors must provide the anti-fibrinolytic strategy used during the study period as it could influence the main outcome...

Answer

Your comment is well-taken.  Of course, the antifibrinolytic strategy could incluence outcomes, especially, if aprotinine is used.  In the current analysis, all three centers used tranexamic acid as the antifibrinolytic strategy, which is especially in Germany common sense.  Nevertheless, this point is extremely valid and therefore, this has been now added into the revised version of the material and methods part.

“Intraoperatively, in all patients an antifibrinolytic strategy with tranexamic acid was applied.”

Question

  1. only 2 patients presented a massive bleeding, it could be interesting to explain why. Did they have any clinical specificity ?

Answer

We greatly appreciated this comment.  We went back into our dataset and could not observed any clinical, preoperative, or other conditions being causative.  Only these 2 patients in the control group experienced a massive bleeding with the only difference of having had no intraoperative hemoadsorption. As the overall number of patients is still low (however being the largest evaluation world-wide so far), we do not want to put more emphasis on that as we just want to present our initial pilot experience.  On the other hand, we totally agree, a potential explanation for this finding (if not the hemoadsorption device) would be of interest.  According to our data, the only difference in these two patients was the application of intraoperative hemoadsorption.  Moreover, as the reviewer suggested, we included the power calculation, which exhibited, that the “treatment” must come with a strong power to result in such large effects.  This interesting finding is in line with the referenced study from Oslo, who could show that even after 15 minutes of bench-top circulation nearly a complete removal of apixaban could be observed.

Question

  1. Idarucizumab and andexanet alfa are presented in the discussion, should they be discussed in the introduction also?

Answer

We are thankful for this remark.  According to your suggestion, we have now added the following regarding reversal agents into the revised version of the introduction:

“On the other hand, new costly reversal agents (idarucizumab targeting dabigatran and andexanet alfa targeting rivaroxaban and apixaban) are currently investigated.”

Question

  1. It could be pertinent to add a flow shart and discuss the exclusion criteria of "life expectancy <1 year", did you study team had Aortic surgery among patients with DOAC excluded ?

Answer

We appreciate this additional remark.  To clarify, in all three centers (which is common opinion in Germany), all patients with a life expectancy <1 year were excluded routinely from cardiac surgery including CPB.  As a life expectancy implies cancer, in Germany or at least in all three participating sites, no on-pump cardiac surgery is being performed.  This differs of course for “on-invasive” or no-CPB procedures such as PCI or TAVI.  To answer your last comment, no aortic cases have been included.  This has now been clarified in the revised version of the manuscript:

“Exclusion criteria were washout periods >48h, use of CytoSorb® for purposes other than antithrombotic removal, pregnancy, life expectancy < 1 year (potential cancer as an underlying disease) and age < 18 years.”

And

“In addition to this, no patients presenting with aortic disease have been included into the present analysis.”

Reviewer 2 Report

Authors have presented an interesting study to evaluate effectiveness of hemo-absorption in patient undergoing cardiac surgery while on apixaban. They looked at 25 consecutive patient 12 with initial controls and 13 following patients with hemo-absorption. They show that bleeding events, chest tube output was lower in intervention group while use of DDAVP and length of surgery (dry up) was less in the intervention group. 

My questions are as follows:

1- Is there any potential side effects of using the hemo-absorption device? Please discuss.

2- Please indicate in your preoperative characteristics whether patients were on any other anticoagulation or anti-platelet. In other word, was there other bleeding risk factors. 

3- I like the speculation of shorter operative time being the result of shorter dry-up time. But only comparing the operative time is not accurate. Please compare time from discontinuation of CPB to skin closure or at least subtract CPB time from operative time and then compare. This is an interesting finding but needs to be better refined. 

4- Authors discuss the new reversal agents for DOACs. This is potentially the main drawback of hemo-absorption technique. Please expand on this and defend your technique better. For example, what is the cost difference? Mention that hemo-absorption removes some of the anti-platelets agents as well. 

Author Response

Reviewer 2

Authors have presented an interesting study to evaluate effectiveness of hemo-absorption in patient undergoing cardiac surgery while on apixaban. They looked at 25 consecutive patient 12 with initial controls and 13 following patients with hemo-absorption. They show that bleeding events, chest tube output was lower in intervention group while use of DDAVP and length of surgery (dry up) was less in the intervention group.

My questions are as follows:

Question

1- Is there any potential side effects of using the hemo-absorption device? Please discuss.

Answer

Your remark is well-taken.  So far, no side effects are known with the device.  In the European market, it is routinely used for antithrombotic removal and also no adjustment of the intraoperative heparin protocol must be performed.  Of note, no adverse device events have been published so far.

Therefore, we added the following into the revised version of the discussion part:

“So far, no other complications have been reported without any other side-effects.  In addition to this, also the intraoperative heparin regime must not be changed, as the therapy is proven and safe and no heparin removal has been shown.”

Question

2- Please indicate in your preoperative characteristics whether patients were on any other anticoagulation or anti-platelet. In other word, was there other bleeding risk factors.

Answer

You point is well taken and we fully agree that other drugs would also have an impact on bleeding outcomes.  The current cohort was only treated with apixaban preoperatively in regard to DOACs.  Only patients with known coronary heart disease received, in addition to apixaban a daily dose of 100mg aspirin.  This has been added to the revised results part.

“Patients with known coronary heart disease were also treated preoperatively with a daily dose of 100mg aspirin.”

Question

3- I like the speculation of shorter operative time being the result of shorter dry-up time. But only comparing the operative time is not accurate. Please compare time from discontinuation of CPB to skin closure or at least subtract CPB time from operative time and then compare. This is an interesting finding but needs to be better refined.

Answer

We totally agree with this comment.  We already stated this information in the results part: The total operation time (skin-to-skin) was shorter in the HA-group without reaching statistical significance (279.8±56.0 vs. 305.2±76.9min., p=0.35).  We totally agree, that the time needed for hemostasis should be lower.  As we did not measure the time from discontinuation from CPB to skin-closure, we could only present the overall surgical skin-to-skin time.  On the one hand, we like the idea of the reviewer to just subtract the CPB time from operative time, but also the time prior to CPB connection might be important, as also during this period some initial hemostasis must be performed.  Anyway, as we could not present the time from CPB-disconnection to skin-closure, we followed your suggestion and have now added the difference including p-value of both times.

Therefore, the following has been added into the revised version of the results part:

“However, regarding hemostasis, the dry-up time (skin-to-skin time subtracted by CPB-time) was lower in the HA-group (160.1±53.4 vs. 196.1±54.7min., p=0.11) without being significant different.”

And the following was also added into the revised version of the discussion part to put more emphasis on this important topic:

“In addition to this, once the CPB-time was subtracted from the overall operation time, an insignificant lower dry-up time for the HA-group was observed.”

Question

4- Authors discuss the new reversal agents for DOACs. This is potentially the main drawback of hemo-absorption technique. Please expand on this and defend your technique better. For example, what is the cost difference? Mention that hemo-absorption removes some of the anti-platelets agents as well.

Answer

We greatly appreciated this well-founded and well-informed remark.  We agree that some new reversal agents might be a potential drawback, on the other hand, however, some side effects on heparin have been described and as you already mentioned, intraoperative hemoadsorption is by far more cost efficient than these newly and highly-expensive drugs.

According to your suggestion we added the following into the revised version of the discussion part:

“They also come at a very high cost (Andexanet alfa 24,750$, incompatible with heparin and potential rebound effect).  Due to cost effectiveness, intraoperative hemoadsorption has been recently recommended by the National Institute for Health and Care Excellence in UK.  Moreover, Javanbakht et al. could show a cost reduction in a UK-based cost-utility analysis of CytoSorb® in patients treated with ticagrelor [20].  Their analysis revealed that the treatment with CytoSorb® could save almost £4,000 over a 30-day time horizon due to consumption of less blood products, fewer re-thoracotomies, and shorter length of stay in the emergency surgical setting, with the price of the device taken into account [2, 20].”

And:

“Moreover, it has been shown in the past, that hemoadsorption also is capable to remove ticagrelor from blood and is highly cost-effective compared to expensive reversal agents.”

Round 2

Reviewer 1 Report

The current revised manuscript was modified regarding our comments and suggestions.